# Digital Health in Schools: A Systematic Review

Cíntia França [1,2], Francisco Santos [1], Francisco Martins [1,2], Helder Lopes [1], Bruna Gouveia [2,3,4,5], Frederica Gonçalves [2,6], Pedro Campos [2,7], Adilson Marques [8,9], Andreas Ihle [3,10,11], Tatiana Gonçalves [12] and Élvio Rúbio Gouveia [1,2,3,*]

[1] Department of Physical Education and Sport, University of Madeira, 9020-105 Funchal, Portugal
[2] LARSYS, Interactive Technologies Institute, 9020-105 Funchal, Portugal
[3] Center for the Interdisciplinary Study of Gerontology and Vulnerability, University of Geneva, 1205 Geneva, Switzerland
[4] Regional Directorate of Health, Secretary of Health of the Autonomous Region of Madeira, 9004-515 Funchal, Portugal
[5] Saint Joseph of Cluny Higher School of Nursing, 9050-535 Funchal, Portugal
[6] Polytechnic School of Technologies and Management, University of Madeira, 9020-105 Funchal, Portugal
[7] Department of Informatics Engineering and Interactive Media Design, University of Madeira, 9020-105 Funchal, Portugal
[8] CIPER, Faculty of Human Kinetics, University of Lisbon, 1495-751 Lisbon, Portugal
[9] ISAMB, Faculty of Medicine, University of Lisbon, 1649-020 Lisbon, Portugal
[10] Department of Psychology, University of Geneva, 1205 Geneva, Switzerland
[11] Swiss National Centre of Competence in Research LIVES—Overcoming Vulnerability: Life Course Perspectives, 1015 Lausanne, Switzerland
[12] SH SeeHealth, 9050-021 Funchal, Portugal
* Correspondence: erubiog@staff.uma.pt

**Abstract:** Worldwide, the growing digitalization process and increase in smartphone usage have contributed to promoting mobile health (mHealth) services. This study provides an overview of the research targeting the effectiveness of mHealth interventions among children and adolescents in the school environment. A systematic literature review was performed following Preferred Reporting Items for Systematic Reviews and Meta-Analyses (PRISMA) in the PubMed, Web of Science, and Scopus databases. The results show that physical activity and nutrition are the main intervention topics. Health literacy, mental health, productive health, vaccination rates, and social interaction were also considered in mHealth interventions. Of the 13 studies that remained for analysis, 12 described positive outcomes in at least one health variable after using an mHealth tool. Overall, interventions ranged between four and 24 weeks. Only seven studies managed to have at least 80% of the participants from the baseline until completion. Adding personal information, user interaction, and self-reference comparisons of performance seems crucial for designing successful health digital tools for school-aged children and adolescents.

**Keywords:** mHealth; mobile; children; adolescent; eHealth; physical activity; nutrition; social

## 1. Introduction

The growing digitalization process in health care and the increase in smartphone usage have contributed to the development of digital health [1]. Digital health is described as the integration of technologies into healthcare [2], which comprises mobile health (mHealth) services [1]. This includes simple apps and complex technologies designed for patient monitoring and education, improving access to health care services and treatment adherence, and chronic disease management [3,4]. Overall, mHealth has evolved over the past decade with regard to the capacity and usability of mobile devices employed, addressed health conditions, and its overall purpose [5]. However, most of the solutions available are conceived for adult usage [5–7], and details are lacking on the use and effectiveness of these solutions among youth populations. Moreover, in the overall existing

models, there is still the need to adapt digital solutions to different areas considering specific contexts and to narrow the gap between health authorities, users, and mHealth developers [8].

Integrating digital technologies into daily living environments such as schools and healthcare facilities has attracted empirical research [9–11]. Interventions based on promoting healthy lifestyles, such as diet advice and monitoring physical activity (PA) levels, have been continuously growing. Although behavior change is frequently mentioned, the literature has described the need for an adequate description of the interventions and an integration of behavior change techniques as a critical aspect of effective mHealth interventions [12].

According to the literature, addressing children's and adolescents' health literacy is crucial for sustainable development and healthy lifestyle promotion throughout life [13,14]. Children and adolescents spend more time in schools than in any other place except at home [15]. Therefore, schools are a vast platform for enhancing health literacy among pupils and educators [16,17]. Despite the potential of apps for pediatric health change behavior, this is still a largely unexplored topic. Previous research on mHealth services has focused chiefly on privileged adults or the general population, particularly regarding PA self-monitoring and goal setting [18–20]. Thus, the novel aim of this review was to provide an overview of the research targeting the effectiveness of mHealth interventions among children and adolescents in the school environment.

## 2. Materials and Methods

### 2.1. Study Design

The current systematic review was undertaken following the Preferred Items for Systematic Reviews and Meta-Analyses (PRISMA) statement on the transparent reporting of systematic reviews [21]. The study protocol was registered with PROSPERO (CRD42022349149).

### 2.2. Search Strategy

In May 2022, the lead author systematically reviewed three electronic databases (PubMed, Web of Science, and Scopus) to find articles that investigated digital health promotion in the school environment among children and adolescents. Primary source articles published in peer-reviewed scientific journals in the past 10 years and up to 31 May were considered eligible. The search terms used for this review were constructed using the PICOS framework: (1) population were children and adolescents of both genders, aged between 12 to 19 years old, (2) interventions that used digital platforms to monitor any type of health condition or to promote health in schools, (3) any type of comparison pre- and post-intervention, (4) healthy lifestyles outcomes, (5) observational and experimental studies, and (6) articles written in English, Spanish or Portuguese. The terms presented in Table 1 were searched in the title/abstract level, in the three databases selected, and combined with the Boolean operators "OR" and "AND".

**Table 1.** Search terms and keywords used in the search strategy.

| Key Search Terms | Related Search Terms |
|---|---|
| Children and Adolescents | Children OR Adolescent OR youth |
| Infectious Disease | COVID-19 OR Infectious OR Disease * |
| Intervention | Intervention * OR Program * OR Protocol * OR RCT OR "Randomized controlled trial" OR "Experimental" |
| mHealth | mHealth OR "Mobile health" OR eHealth |
| School | School |

### 2.3. Screening Strategy and Study Selection

All returning studies were aggregated and exported into a reference manager (End-Note X20, Thomson Reuters, Philadelphia, PA, USA) for additional assessment once the search was completed. After deleting duplicate entries from the database search, three authors independently reviewed the title and abstract for eligibility (CF, FS, FM). The same authors read all eligible records before determining what studies should be included, and the inclusion and exclusion decisions were made by consensus.

### 2.4. Data Extraction and Harmonization

Data extraction and harmonization were carried out by three authors (CF, FS, FM) using a standardized approach with a consensus including six items: (1) general information (authors name and year of study publication), (2) sample characteristics, (3) study purpose, (4) procedures, (5) measures and instruments, and (6) results.

### 2.5. Study Quality and Risk of Bias

The Effective Public Health Practice Project (EPHPP) was used to assess study quality [22]. The six elements of this instrument that examine selection bias include study design, confounding variables, data collecting methods/instruments, whether the evaluators and participants were "blinded," reports of withdrawals, and dropouts. Based on the predetermined criteria, each category was given a poor, moderate, or high score (Table 2). Three authors performed this process separately (CF, FS, FM). The differences were discussed and resolved by consensus.

**Table 2.** Studies methodological quality assessment using the EPHPP.

| Authors | Selection Bias | Design | Confounders | Blinding | Data Collection Methods | Withdrawals and Dropouts | Overall |
|---|---|---|---|---|---|---|---|
| [23] | Strong | Moderate | NA | Weak | Strong | Strong | Moderate |
| [24] | Weak | Moderate | Strong | Weak | Strong | Strong | Weak |
| [25] | Moderate | Moderate | Strong | Weak | Strong | Strong | Moderate |
| [26] | Weak | Moderate | NA | Weak | Strong | Strong | Weak |
| [27] | Moderate | Strong | Strong | Strong | Strong | Weak | Moderate |
| [28] | Weak | Moderate | NA | Weak | Strong | Strong | Weak |
| [29] | Moderate | Strong | Strong | Weak | Strong | Moderate | Moderate |
| [30] | Moderate | Moderate | Strong | Weak | Strong | Strong | Moderate |
| [31] | Weak | Moderate | NA | Weak | Strong | Strong | Weak |
| [32] | Weak | Moderate | NA | Weak | Strong | NA | Weak |
| [33] | Weak | Strong | Strong | Weak | Strong | Weak | Weak |
| [34] | Moderate | Strong | Weak | Weak | Strong | Weak | Weak |
| [35] | Moderate | Strong | Strong | Weak | Strong | Weak | Weak |

NA (not applicable).

## 3. Results

### 3.1. Study Selection

Figure 1 shows the flowchart of the study selection procedure. A total of 543 articles were identified through the search carried out in the aforementioned databases. Of those, 65 articles were duplicates, and 478 remained for eligibility after their removal. In the next step, the title and abstract screening phase, 405 articles were eliminated. Finally, 73 articles were fully assessed, and 13 were chosen as pertinent for inclusion.

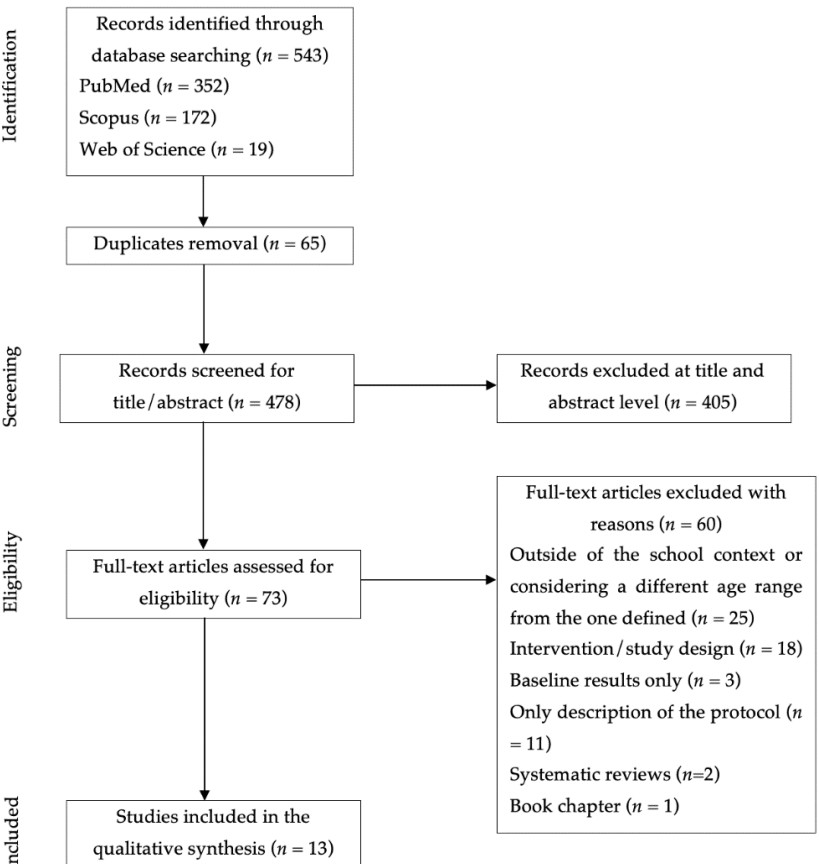

**Figure 1.** Flowchart of the study selection process.

### 3.2. Study Quality and Risk of Bias

The study quality assessment is presented in Table 2. Concerning methodological quality, none of the articles included was classified as strong, five obtained moderate classification [23,25,27,29,30], and eight were poor quality [24,26,31–35]. Considering the instruments categories, it was verified that: (1) only one study was classified as strong in the selection of bias parameter [23], since the rate of participants exceed the 80%, which may be representative of the target population; (2) the randomized controlled trials or controlled clinical trials were classified as strong study design (*n* = 5) [27,29,33–35], while other type of design was classified as moderate (*n* = 8) [23–26,28,30–32]; (3) seven studies revealed no baseline differences between groups in the confounders' section or accounted for at least 80% of significant confounders [24,25,27,29,30,33,35], whereas the studies performed with only one group were not evaluated at this point (*n* = 5) [23,26,28,31,32]; (4) only one study blinded the assessors and participants, being classified as strong [27]; (5) all studies presented valid and reliable data collection instruments; and (6) in the withdrawals and dropouts parameter, studies were classified as strong if the dropout rate did not overcome 20% of the participants (*n* = 7) [23–26,28,30,31], as moderate if the dropout rate was between 60 and 79% (*n* = 1) [29], and as week of the dropout rate was greater than 60% (*n* = 4) [27,33–35].

### 3.3. Intervention Characteristics

The characteristics of each study included in our review are summarized in Table 3. The interventions involved a total of 2757 students. The students' age ranged from 9 to 18 years. Six studies included participants over 15 years [23,25,27,31,32,34,35], and five studies included children below 13 years [24,26,28–30,35]. One investigation has also included a sample of 23 teachers [33].

**Table 3.** Summary of the study description and variables considered.

| Author (Year) | Sample | Purpose | Health Area | Intervention | Instruments and Measures | Main Results |
|---|---|---|---|---|---|---|
| [23] | 400 female students aged between 14 and 19 years | To examine the effect of the mHealth tool on knowledge regarding reproductive health, and to determine the use of the mHealth tool among schoolgirls. | Reproductive health | Short message service (SMS) delivered through a mobile phone for 8 weeks. Each user received three SMS every week. The first one contained a multiple-choice quiz question on reproductive health. The users answered the quiz by replying to the SMS. Then, two SMS with the correct answer and additional information on the quiz topic were sent sequentially. | Reproductive health knowledge (mobile health tool, SMS). | Post-intervention knowledge scores on reproductive health were significantly higher compared to baseline scores. The SMS approach was an easy and effective way to improve reproductive health knowledge for adolescent girls. |
| [24] | 230 students aged between 11 and 15 years (IG = 139; CG = 91) | To assess the feasibility of an eHealth solution to address eating habits and PA in adolescents. | Physical activity and nutrition | Use of an eHealth application ("SanoyFeliz") for 14 weeks. Through the platform, students could connect and interact with other users, post comments and photos, receive notifications and information about nutrition and PA daily, and get virtual rewards for improving their habits. | Eating habits and PA were assessed through the KIDMED (nutrition) and PAQ-A questionnaires. Body measurements (body mass and height) were used to obtain the children's BMI. | A significant statistical decrease in the BMI was observed in the IG for individuals with an initial percentile greater than 50%. KIDMED scores were significantly better in the IG after 14 weeks compared with the CG. However, PA levels have slightly decreased after the intervention, particularly in the CG. |
| [25] | 863 students aged between 14 and 17 years | To improve vaccination rates and knowledge and self-efficacy in a school context. | Vaccination | For 1 week, educational units (about 45 m) were taught by a physician in the classrooms. Information was distributed using a PowerPoint presentation with interactive elements, and social media elements such as newspaper articles and videos were included. All four participating schools received a visit from the Prevention Bus, which contained a medical team, two physicians, two nurses, and a bus driver. | Seven indicators and measures were assessed: school recruitment log, vaccination documents, vaccinations delivered, rating of the educational unit, semistructured interviews on the education unit, vaccination-related knowledge scale, and vaccination-related perceived self-efficacy scale. | From the whole sample, 437 students (50.9%) brought their vaccination cards to a school, and 79 received vaccinations. Students improved their scores in six of the knowledge questions in the post-intervention. The teaching methods (interactive and social media components) were perceived as very good by the participants. |

**Table 3.** *Cont.*

| Author (Year) | Sample | Purpose | Health Area | Intervention | Instruments and Measures | Main Results |
|---|---|---|---|---|---|---|
| [26] | 102 students (39 females and 63 males) aged 9 and 12 years | To describe the development and user testing of a nutrition education gamified app designed for children. | Nutrition | Testing a gamified mHealth app ("Foodbot Factory") designed to improve food and nutrition knowledge among children. Five interactive user testing sessions were conducted for approximately 20 to 30 min using an Apple iPad (iOS 12) or a Lenovo tablet (Android 8.1.0). | Qualitative interviews and questionnaires to assess users' satisfaction, engagement, usability, and knowledge gained. | In the final user test, most users still found the app was fun, had clear goals, and was easy to use (71–94%). A total of 71% of students have shown interest in still using the Footbot Factory after the intervention. |
| [27] | 41 students aged 15.6 ± 0.25 years (IG = 20; CG = 21) | To describe the usage and feasibility of an mHealth intervention concerning self-efficacy levels, and emotional and physical health. | Physical activity, nutrition, and mental health | The IG had access to the app for 6 weeks, and measurements were compared between this time point and the baseline. Multiple focus group studies were performed among adolescents and advisors for app development. Based on the results, the app was built as a social health game. Through gamification, the app functionality aims to help users set goals and develop health-related missions in three main categories: nutrition, PA, and mental health. | The app's acceptability and functionality were assessed with the Systematic Usability Scale. Further, the amount, frequency, ad time of daily PA was measured through in-app activity. The stress levels, quality of sleep, and energy levels were evaluated by completing in-app health tasks. The anthropometric assessment included height and weight. | The reported daily PA increased by nearly 20% in the IG, dropping by almost 26% in the CG. Self-efficacy levels increased by 8% in the IG and decreased by 3% among the CG. |
| [28] | 24 students aged between 12 and 14 years | To investigate whether a technology-based educational program that combines education, PA, and self-assessment of goal achievement, would contribute to changing PA behaviors toward the international PA recommendations. | Physical activity and nutrition | The iEngage educational program was implemented through an app for 4 weeks. The app targeted health literacy, PA-related skills, and sugar-focused nutrition guidelines. Learning activities, goal setting, self-assessment tasks, and brief 2- to 5-min PA sessions (focused on particular movements such as sprints, squats, jumping, etc.) were developed in two modules of 1 h per week. | Anthropometric and physical fitness data (aerobic capacity, speed, and agility) were assessed three weeks before the program. The baseline PA behavior was measured using research-grade activity sensors (GENEActiv) five consecutive days before the program. PA during the program was measured using Misfit activity trackers. | On average, participants achieved 11197 ± 1376 steps per day during the 4-week intervention. PA showed an overall increase, particularly in the less active individuals (an increase of nearly 15% in the daily steps). The satisfaction with the modules was 95% across the program |

**Table 3.** *Cont.*

| Author (Year) | Sample | Purpose | Health Area | Intervention | Instruments and Measures | Main Results |
|---|---|---|---|---|---|---|
| [29] | 125 students aged between 9 and 13 years (IG1: MARA = 2 classes; IG2: MARA + SMS = 2 classes; CG: 3 classes) | Assess the potential cohesion effect of a PA school-based intervention by analyzing longitudinally the friendship network structure and the mechanisms of friendship formation/dissolution. | Physical activity and social interaction | The interventions were implemented for 10 weeks. The IG1 (MARA group) was intervened three times per week during the school recess of 30 min. In total, 30 sessions of PA combined with supervised games with ties, balls, hoops, stairs, and dancing were carried out. The IG2 (MARA + SMS group), in addition to the intervention made in IG1, was also targeted with SMS each weekday. The SMS was focused on promoting the students' participation, engagement, motivation, and empowerment, in extracurricular PA and healthy behaviors among their classmates and family members. The CG did not receive any intervention. | Socioeconomic status was assessed using the demographic and family health questionnaire. The health-related assessment included anthropometry (body mass and height) and accelerometry (GT3X+ accelerometer) to provide BMI. The network structure and cohesion information were collected using an interview. Users' satisfaction levels were evaluated using the PA class satisfaction questionnaire). | The intervention influenced the mechanisms of friendship formation and dissolution. On average, the MARA + SMS group showed more social cohesion and 3.8 more friendships than the program alone. PA levels and BMI began to affect friendship homophily and the formation/dissolution of friendships in the intervened networks over time. Children became more likely to stop being friends with their peers with different BMIs (network 1 of MARA + SMS). Besides, children became less likely to become friends with their peers with varying levels of PA (network 2 of MARA + SMS and network 3 of MARA) |
| [27] | 126 students aged between 9 and 13 years | To test the effectiveness of an intervention to increase students' digital health literacy and health knowledge. | Health literacy, physical activity, nutrition, and social interaction | Teachers delivered a classroom-based education program (Learning for Life) over 6 weeks in three schools. Teachers were provided with an educator's toolkit, student workbook, and online interactive graphics for students focused on health literacy PA, sedentary behavior, and social connectedness. | Technology usage was assessed by self-reported past and current internet usage and which devices they use. Digital health literacy was evaluated using the eHealth Literacy Scale. The interview collected PA levels and sedentary behavior, including the amount of time spent doing sedentary activities. Health knowledge and behavior change included a 10-item questionnaire. | From pre- to post-intervention, students' digital health literacy improved. However, there was a significant decrease in digital health literacy from post-intervention to follow-up (2 months after the intervention). In the post-intervention assessment, most students could identify at least one healthy behavior (e.g., exercising one hour per day) and reported making at least one healthy change in their lives (e.g., eating more fruits or vegetables). |

**Table 3.** *Cont.*

| Author (Year) | Sample | Purpose | Health Area | Intervention | Instruments and Measures | Main Results |
|---|---|---|---|---|---|---|
| [31] | 33 students aged between 16 and 18 years (24 females and 9 males) | Test the feasibility of a mobile application and examine whether it could be used to monitor dietary intake among adolescents. | Nutrition | Three-month intervention study. Participants answered pre- and post-intervention dietary habit questionnaires. Participants were asked to record all foods and beverages consumed using voice or text input. Nutrient intake was measured using 24 h recalls pre-pre-and-intervention. | Monitor Dietary intake (Moblie App, "Diet-A"). | There was a significant decrease in sodium and calcium intake between pre-and post-intervention. Nearly 61.9% of the participants reported being satisfied with the app's usage to monitor their food intake, and 47.7% liked getting personal information about their dietary intake. However, more than 70% answered that using the app was burdensome or had trouble remembering to record their food intake. |
| [32] | 42 students aged between 15 and 18 years | Develop and assess the acceptability of an avatar-based mobile app (Monitor Your Avatar, MYA). | Self-perception | Cross-sectional study. First, the research team measured height, weight, body fat percentage, and adolescents' body parts. Then, it was divided into three phases: (i) Perceived Avatar—adolescents designed the avatar to represent how they currently perceive their bodies to look; (ii) Target Avatar—adolescents take the first avatar and transform it as they want their bodies to look within realistic, healthy parameters; (iii) Actual Avatar—adolescents enter their body part measurements into the app and generate an actual avatar of themselves. | Interactive and designed mobile health app (MYA). Open-ended reaction questions to assess participants' acceptability to MYA. | Avatar-based mobile apps provide immediate feedback and allow users to engage with personalized images to represent their perceptions and actual body images. The participants reacted positively to the avatar app and could view avatars that represented them. |

**Table 3.** *Cont.*

| Author (Year) | Sample | Purpose | Health Area | Intervention | Instruments and Measures | Main Results |
|---|---|---|---|---|---|---|
| [33] | 313 participants (290 students aged between 11 and 13 years, and 23 teachers) | Evaluate how social comparison drives preadolescents' engagement with an mHealth app. | Physical activity, nutrition, and social interaction | Students and teachers used an mHealth tool ("GameBus") for 12 weeks that rewarded healthy activities. Three different social comparative settings as treatments, test whether an intergroup competition would be more effective in promoting healthy habits. A crossover study design was adopted to ensure that all the participants were exposed twice to every treatment. Each treatment simulated a different implementation of the social comparison technique: (i) intragroup competition, (ii) intergroup competition, and (iii) intergroup competition, increasing teachers as potential role models for students. | Use an mHealth tool for health promotion (mobile app, "GameBus"). Post-intervention survey (individual factor proposed by the social comparison model of competition; students' perception of closeness to their peers, similarity to their teachers and/or peers, the relevance of the prescribed activities, and personality). | An intergroup competition can increase preadolescents' passive engagement with mHealth apps. However, an intergroup competition does not necessarily result in preadolescents performing more unique activities on average. The active involvement of a role model (e.g., a teacher) can influence the average number of activities performed by preadolescents in an intergroup setting. |
| [34] | 105 students aged between 16 and 18 years (IG = N/S; CG = N/S) | Encourage high school students to meet PA goals using a newly developed game. | Physical activity | Twelve-week pilot test. Students were randomly assigned to a Game Condition or Control Condition. The difference was that the Game Condition received access to the Camp Conquer game, and the number of steps and active minutes was translated into coins and gems. | The number of steps and the amount of time spent in activity per day were collected using the Fitbit devices. Additional data included the number of logins into the gaming platform and a baseline questionnaire focused on PA, sleep, gaming, and dietary patterns. | The intervention was not successful in increasing PA in high school students. Nearly 50% of the participants did not consistently wear their FitBit or engage in the gaming intervention. Logistical factors, such as needing to charge the FitBit and take it off for sports/sleep, and game glitches, were some of the reasons. |

**Table 3.** *Cont.*

| Author (Year) | Sample | Purpose | Health Area | Intervention | Instruments and Measures | Main Results |
|---|---|---|---|---|---|---|
| [35] | 353 students aged between 12 and 16 years (IG = 140; CG = 213). | Evaluate the effectiveness of lifestyle change of an mHealth intervention to promote healthy behaviors in adolescence (TeenPower). | Health literacy | For 6 months, the IG was invited to engage in the mHealth intervention (TeenPower) in addition to school-based intervention. The CG followed the school-based intervention (face-to-face psycho-educative sessions with nutritional, behavioral, and PA counseling). The IG was access to the TeenPower mobile software application: created for adolescents to provide them with educational resources, social support, self-monitoring features, interactive training modules, and motivational tools. | A questionnaire assessed the Adolescent Lifestyle Profile, which included information on health responsibility, PA level, nutrition, positive life perspective, interpersonal relationship, stress management, and spiritual health. Body image dissatisfaction. The eHealth literacy was evaluated using the eHealth Literacy Scale tool. The Body Image perception was measured using a sequence of seven silhouettes and the presently estimated body dissatisfaction minus the desired body shape. | The post-intervention assessment dropped significantly (IG = 53 and CG = 151 students). Although the considerable dropout rate, mHealth intervention (TeenPower) significantly affects nutrition, positive life perspective, and global lifestyle outcomes. |

IG (intervention group), CG (control group), N/S (non-specific), PA (physical activity), BMI (body mass index), BF% (body fat percentage), PE (physical education).

The study's duration is displayed in Figure 2. Four studies evaluated a digital tool for four weeks or less [25,26,28,32], three studies lasted between four and eight weeks [23,27,30], while the majority varied from eight to 16 weeks [24,29,31,33,34]. The longest intervention was performed for 24 weeks [35].

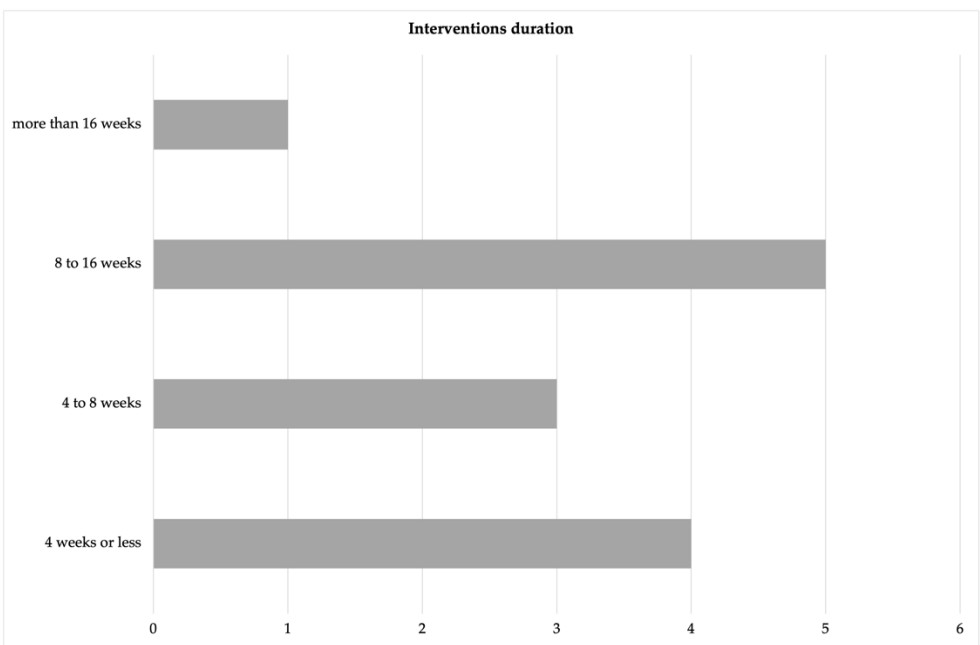

**Figure 2.** Summary of the results regarding intervention duration.

The instruments used during interventions are presented in Figure 3. Most interventions were performed through a mobile app [24,26–28,31–35]. Two programs used short message services (SMS) [23,29], and another three primarily relied on educational units during school time [25,29,30].

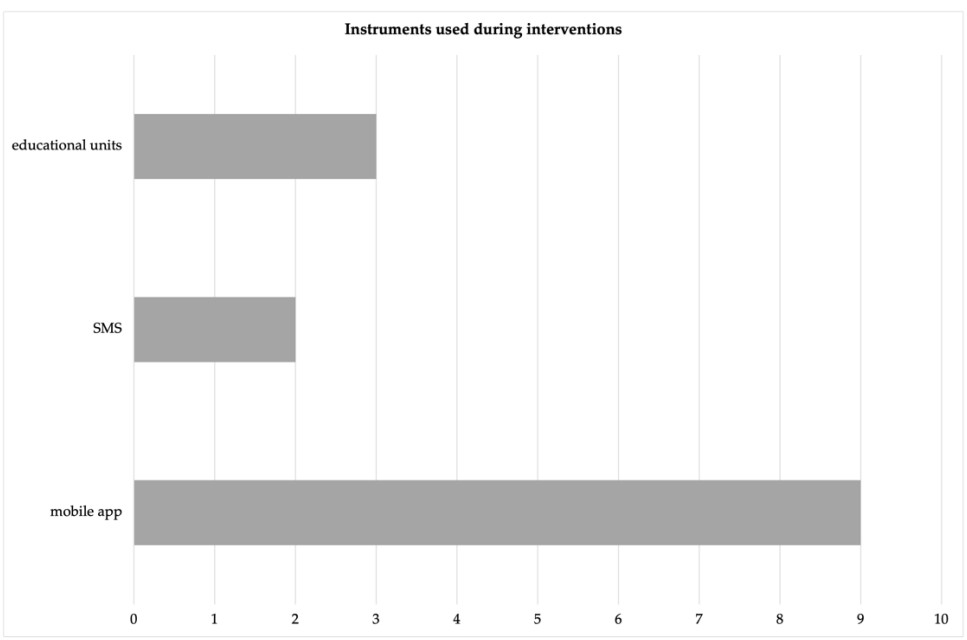

**Figure 3.** Summary of the mHealth instruments used in interventions.

*3.4. Main Results*

Table 3 shows the study description and the variables considered.

The post-intervention assessment dropped significantly (IG = 53 and CG = 151 students). Although the considerable dropout rate, mHealth intervention (TeenPower) significantly affects nutrition, positive life perspective, and global lifestyle outcomes.

Different health areas were covered (Figure 4), particularly PA and nutrition [24,26–31,34]. Besides, mental health [27], reproductive health [23], vaccination [23], social interaction [29,30,33], overall health literacy [30,35], and self-perception [32], were topics covered in the articles retained for analysis.

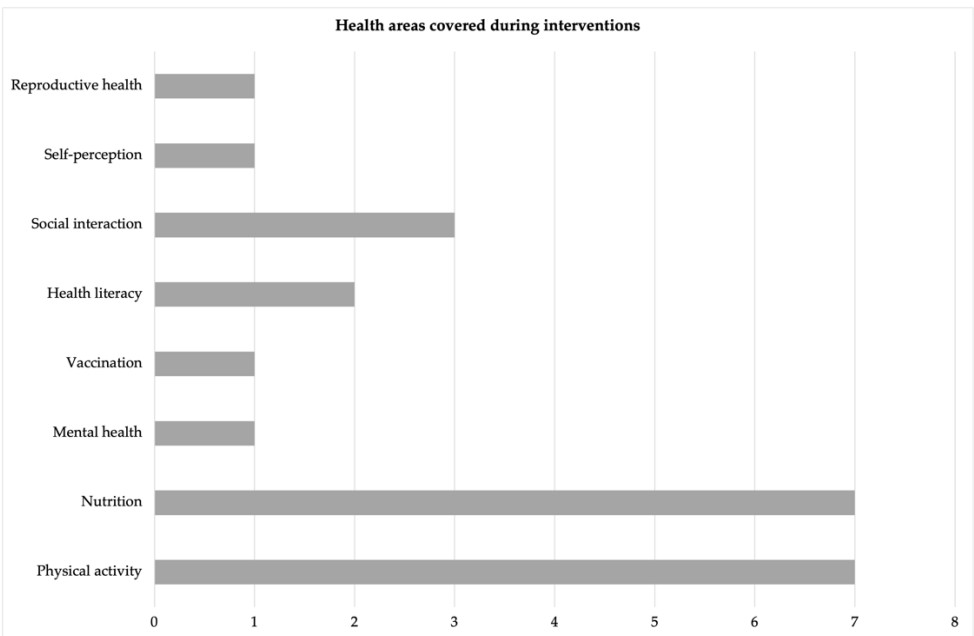

**Figure 4.** Summary of the health areas covered in the studies analyzed.

Regarding PA, the positive effects of mHealth were reported in four studies [27–30], while one investigation did not describe a positive impact of a gamified approach on increasing adolescents' PA levels [34]. Regarding nutrition, six interventions pointed out positive outcomes [24,26–28,30,31]. One of the previous studies combined PA and nutrition interventions, reporting positive outcomes in nutrition but not in PA levels [24].

On the other hand, five studies reported a positive impact of mHealth interventions focused on social interaction, mental health, and self-perception [27,29,30,32,33]. Another four studies described mHealth usage as beneficial to improve reproductive health, vaccination rates, and overall health literacy [23,25,30,35]. Besides students, one study also involved teachers in evaluating how social comparison drives preadolescents' engagement with an mHealth app, concluding that the involvement of a role model (e.g., a teacher) can influence the average number of activities that preadolescents perform in an intergroup setting [33].

## 4. Discussion

This study aimed to perform a systematic review of the literature targeting the effectiveness of mHealth interventions among children and adolescents in the school environment. Overall, interventions targeted students' PA levels, nutrition, and general health components. Of the 13 studies that remained for analysis, 12 described positive outcomes in at least one of the previous components mentioned after using an mHealth tool [23–33,35]. In contrast, one intervention did not successfully increase PA in high school students aged between 16 and 18 years [34].

PA was the focus of most interventions, although most combined it with nutrition [24,26–28]. Indeed, overweight and obesity prevalence rates among children and adolescents are an alarming and increasing problem [36], which could be changed by avoiding sedentary behavior and promoting a healthy diet. Most of the interventions focused on this topic were promoted through apps targeting PA-related skills, gamified health-related situations, and nutritional guidelines [24,27,28]. Positive outcomes were reported regarding the number of steps per day [28], time spent in daily PA [27], and BMI assessment [24]. In a study conducted for six weeks among adolescents aged 15.6 $\pm$ 0.3 years, the daily reported PA levels increased nearly 20% in the IG, dropping by 26% in the CG. However, a significant decrease was observed in average exercises performed between the first week and the subsequent intervention [27]. Another four-week investigation concluded an improvement of nearly 27% in PA behaviors over the program, particularly for adolescents who were least active at the baseline. Interestingly, these adolescents more often met the international recommendations for daily steps and PA intensity by the end of the program than their more active peers [28].

On the contrary, in a sample of adolescents aged 16 to 18 years, a 12-week gamified intervention that allowed students to get game rewards through their PA levels was unsuccessful. The IG did not present more steps or active minutes than the CG (assessed using Fitbit). Moreover, 50% of the students participated for fewer than 10 days, and only 21 individuals out of the 105 initial sample played at least one game [34]. The authors reported using Fitbit as a constraint since most participants did not use it during the intervention. Another study using Fitbits for eight weeks also described the lack of consistency in wearing these devices and the decreased motivation for PA in an adolescent population [37]. One possible explanation for these outcomes was that participants felt less competent when they did not reach the 10,000 steps/day goal and because they felt like competing with their friends [37]. Therefore, it is recommended to encourage self-referenced comparisons of performance instead of engaging in normative comparisons with peers or established recommendations.

Research among school-age children indicates that they may be significantly influenced by their friends' PA levels and obesity-related behaviors [38,39]. Literature has mentioned that physically active children positively impact their peers' PA [40–42]. The investigation conducted on 125 students between 9 and 13 years concluded that PA and BMI affected friendship formation and dissolution. In summary, children became more likely to stop being friends with children with different BMI or PA levels and more likely to relate with others with an affinity for PA or BMI [29]. Indeed, shared common interests should be one of the most decisive factors for friendship and choosing to spend time with others during recess.

Meanwhile, eating habits were significantly improved after 14 weeks of mHealth app usage in 139 children aged between 11 and 15 years [24]. Another study aimed to monitor dietary intake through an app in 33 students aged between 16 and 18 years, reporting a significant decrease in sodium and calcium intake between pre-and post-intervention analyses [31]. However, more than 70% of the participants reported trouble remembering to record their food intake [31]. Although some real advantages emerge from mHealth tools over paper diaries, various challenges and obstacles in food intake monitoring still exist, particularly concerning its accuracy [43]. A critical finding in the study of Lee et al. [31] was that nearly 48% of the participants liked getting personal information about their dietary intake from the app, which may underline the value of individualized approaches while using mHealth solutions.

In this review, only one study has considered teachers in the sample to assess social engagement with an mHealth app [33]. The results showed that the active involvement of a role model (i.e., a teacher) could impact the number and type of activities that preadolescents perform. Additionally, the authors observed that students monitored the intergroup competition more closely than the intragroup competition since they checked the app more often when involved in team-based comparisons [33]. Understanding the students' moti-

vations is crucial to establish the requisites of an mHealth solution focused on promoting healthy lifestyles. Otherwise, the use of the digital tool may be restricted at the beginning of the intervention when it is still a novelty. In fact, in this review, only seven studies managed to get 80–100% of the participants from the baseline to the intervention completion. In contrast, six studies presented a dropout rate of at least 20%. The highest dropout rate corresponded to nearly 62% in the IG and 30% in the CG among students aged between 12 and 16 years [35].

Digital health tools were also used in the school context to improve vaccination rates and knowledge regarding reproductive health among participants aged between 14 and 19 years [23,25]. In both studies, educational support given by interactive classroom content or SMS proved to be an easy and effective approach to enhance students' awareness and literacy concerning these topics. However, the literature has identified gamified situations as more attractive among children [44,45]. Rewards, feedback, and socialization aspects are frequently employed through gamified mHealth [45]. Therefore, stakeholders should consider the users' age range during the deployment of digital solutions to provide a successful tool.

Although the number of studies included in this review is limited, it underlines an important gap in the literature concerning the use of mHealth designed in the school environment. Childhood and adolescence are crucial for developing healthy lifestyles, such as regular engagement in PA and a healthy diet, and increasing health literacy [38]. Since youngsters are a significant part of their day-involved schools, this can be a privileged context to implement strategies focused on promoting health. Overall, mHealth tools have effectively changed short-term behaviors and increased network cohesion. However, shared challenges have emerged concerning dropout rates or the continuous decrease in users' interaction with the respective tools during interventions. On the other hand, although the growing increase in smartphone usage, it is still important considering youngers with fewer opportunities for technology access.

Our results bring critical practical implications for the future design of mHealth solutions, such as considering the target audience age range, allowing the possibility of user interaction, and including self-referenced performance comparisons instead of focusing exclusively on group comparisons. Furthermore, it could be beneficial to gradually introduce new and appealing content in future digital solutions to avoid app usage decline over time. Future research is still needed to investigate which contents and strategies might be more effective in maintaining youngsters' engagement in mHealth solutions, particularly considering different age ranges and gender.

## 5. Conclusions

Results from this systematic review suggest that mHealth tools are effective for short-term behavior change and developing knowledge towards health. Intervention duration ranged between four and 24 weeks, and only one study did not report positive outcomes from a 12-week pilot study based on a gamified situation. Overall, only seven studies managed to have at least 80% of the participants from the baseline to the intervention completion, and app usage tended to decline post-intervention. Adding personal information, user interaction, and self-reference comparisons of performance seems crucial for designing a successful digital tool for behavior change in school-aged children and adolescents.

**Author Contributions:** Conceptualization, C.F., F.S., F.M., and É.R.G.; methodology, C.F., F.G., and F.M.; software, C.F., F.S., and F.M.; validation, B.G., H.L., F.G., P.C., A.M., T.G., A.I., and É.R.G.; formal analysis, C.F., F.S., and F.M; investigation, C.F., F.S., and F.M.; resources, F.G., P.C., B.G., H.L., T.G., and É.R.G.; data curation, C.F., F.S. and F.M.; writing—original draft preparation, C.F., F.S., and F.M.; writing—review and editing, H.L., B.G., F.G., P.C., A.M., T.G., A.I., and É.R.G.; visualization, A.M., A.I., and É.R.G.; supervision, B.G., A.M., H.L., and É.R.G.; project administration, É.R.G.; funding acquisition, P.C., F.G., H.L., B.G., and É.R.G. All authors have read and agreed to the published version of the manuscript.

**Funding:** É.R.G., C.F., F.M., F.G., P.C., and B.G. acknowledge support from LARSyS—the Portuguese national funding agency for science, research, and technology (FCT) pluriannual funding 2020–2023 (Reference: UIDB/50009/2020). This study is framed in the Saúde Escolas: Projeto de Monitorização em Saúde (SEE_App). The project received funding under application no. M1420-01-0247-FEDER-000042 in the System of Incentives for the Production of Scientific and Technological Knowledge in the Autonomous Region of Madeira—PROCiência 2020.

**Institutional Review Board Statement:** Ethical review and approval were waived for this study since this study comprises a systematic review of digital health in schools. The study protocol was registered with PROSPERO (CRD42022349149).

**Informed Consent Statement:** Not applicable.

**Data Availability Statement:** Not applicable.

**Acknowledgments:** Not applicable.

**Conflicts of Interest:** The authors declare no conflict of interest.

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
