# Peer review of "Digital Health in Schools: A Systematic Review"

_sustainability, doi:10.3390/su142113848_

Round 1

Reviewer 1 Report

Thank you for giving me the opportunity to review the paper. This study provides an overview of the 27 research targeting the effectiveness of mHealth interventions among children and adolescents in the 28 school environment by performing a systematic literature review. Let me suggest some comments as follows:

1.       Both the method and procedure of the study seem appropriate. However, it would be good to present the definition of "Digital health" and the background of this study in the introduction.

2.       Second, I think it will be more meaningful to increase the value of this study if more in-depth discussions are presented on the implications and utilization plans of this study in the Discussion part.

Author Response

Thank you for giving me the opportunity to review the paper. This study provides an overview of the research targeting the effectiveness of mHealth interventions among children and adolescents in the school environment by performing a systematic literature review. Let me suggest some comments as follows:

  1. Both the method and procedure of the study seem appropriate. However, it would be good to present the definition of "Digital health" and the background of this study in the introduction.

Response 1: We would like to thank the reviewer for the overall positive feedback in our manuscript. As suggested, we included a definition of “digital health” in the Introduction section as follows: “Digital health is described as the integration of technologies into healthcare [2], which comprises mobile health (mHealth) services [1].“(line 44-46)

  1. Second, I think it will be more meaningful to increase the value of this study if more in-depth discussions are presented on the implications and utilization plans of this study in the Discussion part.

Response 2: As suggested by the reviewer, we perform a more in-depth discussion on the practical implications of our study at the end of the Discussion section.

Reviewer 2 Report

1-According to the PICOS framework was used in this study, the healthy lifestyles considered as outcomes , but infectious diseases were searched for in the key search terms. It seems that the keywords of infectious diseases are not suitable for healthy lifestyles and the search results in this section cannot be trusted. It would have been better to use the keywords "health OR healthy " and "lifestyle" for searching.

2-Please explain exactly the study environment. In this review, are there any studies that the child used the mobile phone for health issues only at school or both at school and outside of school? Did the child not have the right to use mobile applications outside of school?  The word school environment is ambiguous and needs further explanation. Basically, what is the purpose of the authors regarding the necessity of choosing a school for health? Were they looking to determine the impact of teachers and school officials on the use of mhealth applications or something else?

3- In the discussion section, it is better to mention the factors that can affect the effectiveness of mhealth in school. Do the authors have any suggestions for future research in this regard?

Author Response

1-According to the PICOS framework was used in this study, the healthy lifestyles considered as outcomes , but infectious diseases were searched for in the key search terms. It seems that the keywords of infectious diseases are not suitable for healthy lifestyles and the search results in this section cannot be trusted. It would have been better to use the keywords "health OR healthy " and "lifestyle" for searching.

Response 1: We understand the reviewer’s opinion regarding the search terms. The term “infectious disease” was used since digital health have been massively used to manage infectious diseases such as coronavirus. Besides, infectious diseases epidemics are a concerning for public health, and the use of digital tools could be crucial in tackling focus of infections.

2-Please explain exactly the study environment. In this review, are there any studies that the child used the mobile phone for health issues only at school or both at school and outside of school? Did the child not have the right to use mobile applications outside of school?  The word school environment is ambiguous and needs further explanation. Basically, what is the purpose of the authors regarding the necessity of choosing a school for health? Were they looking to determine the impact of teachers and school officials on the use of mhealth applications or something else?

Response 2: We appreciate the reviewer in-depth analysis. The choice of deepen the knowledge on the use of mHealth solutions in the school environment is sustained by the idea that childhood and adolescence are crucial “windows of opportunity” to promote health literacy and sustainable development throughout life. Since children and adolescents spent a great part of their day in schools, this seems to be a vast platform to enhance health literacy. We have updated the Introduction section to clarify this idea.

The concept of “school environment” refers to all the activities performed inside the school facilities, including classes and recess time.

This study was focused on the use of mHealth exclusively by healthy students. Interventions designed for populations with specific characteristics (e.g., overweight children) or at different age were outside of our scope. The inclusion criteria considered digital solutions that had to be used in schools and could also be used outside the schools.

3- In the discussion section, it is better to mention the factors that can affect the effectiveness of mhealth in school. Do the authors have any suggestions for future research in this regard?

Response 3: As suggested, in the end of the Discussion section, we added information regarding the factors than can affect mHealth usage in schools and we have made suggestions for future research in this topic as follows: “Since youngsters are a significant part of their day-involved schools, this can be a privileged context to implement strategies focused on promoting health. Overall, mHealth tools have effectively changed short-term behaviors and increased network cohesion. However, shared challenges have emerged concerning dropout rates or the continuous decrease in users’ interaction with the respective tools during interventions. On the other hand, although the growing increase in smartphone usage, it is still important considering youngers with fewer opportunities for technology access.

Our results bring critical practical implications for the future design of mHealth solutions, such as considering the target audience age range, allowing the possibility of users’ interaction, and including self-referenced performance comparisons instead of focusing exclusively on group comparisons. Furthermore, it could be beneficial to gradually introduce new and appealing content in future digital solutions to avoid app usage decline over time. Future research is still needed to investigate which contents and strategies might be more effective in maintaining youngsters’ engagement in mHealth solutions, particularly considering different age ranges and gender.” (line 313-328)

Reviewer 3 Report

·         There are a few papers that need to trim out and include the latest papers in order to find the research gaps.

·         In the literature survey, authors should write research gaps in existing models.

·         On what basis, authors have produced the remarks “strong, moderate and weak” in Table 2. Studies methodological quality assessment using the EPHPP. Please provide proper justification.

·        Is there any simulation results to say this “Overall, only seven studies managed to have at least 80% of the participants from the baseline to the intervention 280 completion, and app usage tended to decline post-intervention”. Please provide proper justification.

·         The contributions in the paper should be enhanced and presented clearly. Compared with existing works, what are the advantages of the methods proposed in this paper?

·         Authors have to provide more diagrams about their contribution to this manuscript to attract the target audience.  

·         This manuscript can be accepted after major revision.

Author Response

  1. There are a few papers that need to trim out and include the latest papers in order to find the research gaps.

Response 1: Our data collection considered the search terms and the inclusion criteria presented in the Materials and Methods section. After deleting the duplicates, three authors did the studies revision thoroughly. In situations where there was doubt about the study inclusion/exclusion, the three authors debated and reach a consensual decision. We acknowledge that the sample in our manuscript is, unfortunately, small. However, this should represent an opportunity to promote future research on this topic, since the literature is still scarce (considering the number of studies included for analysis).

  1. In the literature survey, authors should write research gaps in existing models.

Response 2: We thank the reviewer’s feedback. We added some information concerning the gaps in the existing models in the Introduction section as follows: “Moreover, in the overall existing models, there is still the need to adapt digital solutions to different areas considering specific contexts and to narrow the gap between health authorities, users, and mHealth developers [8].” (line 52-55)

  1. On what basis, authors have produced the remarks “strong, moderate and weak” inTable 2. Studies methodological quality assessment using the EPHPP. Please provide proper justification.

Response 3: The EPHPP tool was used to assess studies’ methodological quality. This tool was created to address articles in a wide range of health-related topics and uses pre-determined criteria that guides the investigator during the assessment of each category. Once the assessment is fulfilled, each article examined receives a mark ranging between “strong”, “moderate”, and “weak” in the categories presented in Table 2.

  1. Is there any simulation results to say this “Overall, only seven studies managed to have at least 80% of the participants from the baseline to the intervention 280 completion, and app usage tended to decline post-intervention”. Please provide proper justification.

Response 4: Through the quality assessment tool (EPHPP) the dropout rate was estimated for each study, allowing us to achieve the results mentioned above.

  1. The contributions in the paper should be enhanced and presented clearly. Compared with existing works, what are the advantages of the methods proposed in this paper?

Response 5: To the best of our knowledge, this is the first systematic review focused on mHealth solutions designed for healthy children and adolescents in school context. We added in the Introduction section the main areas covered in previous research conducted in mHealth solutions to better justify our approach.

  1. Authors have to provide more diagrams about their contribution to this manuscript to attract the target audience.

Response 6: We thank the reviewer’s useful suggestion. We added some graphical data to better illustrated our findings which might be consulted in the manuscript (Figure 2, 3, and 4).

Round 2

Reviewer 2 Report

Unfortunately, the authors' response to the comments is not convincing.

Reviewer 3 Report

Authors have updated the manuscript based on suggestions